# Formation Investigation of Intermetallic Compounds of Thick Plate Al/Mg Alloys Joint by Friction Stir Welding

**DOI:** 10.3390/ma12172661

**Published:** 2019-08-21

**Authors:** Yang Xu, Liming Ke, Yuqing Mao, Qiang Liu, Jilin Xie, Haoran Zeng

**Affiliations:** 1State Key Laboratory of Solidification Processing, Northwestern Polytechnical University, Xi’an 710072, China; 2National Defence Key Discipline Laboratory of Light Alloy Processing Science and Technology, Nanchang Hangkong University, Nanchang 330063, China

**Keywords:** friction stir welding, Al/Mg alloys, thick plate, intermetallic compounds

## Abstract

5A06 Aluminum (Al) alloy and AZ31B magnesium (Mg) alloy with 20 mm thickness were successfully butt joined by friction stir welding. In order to control the composition of Al and Mg alloys along thickness direction, an inclined butt joint was designed in this study. The microstructure and phase identification at the interface of Al/Mg joints were examined using scanning electron microscopy with an energy-dispersive spectroscopy and Micro X-ray diffraction. The results indicated that there were two different formation mechanisms of intermetallic compounds at the interface of thick plate Al/Mg joint. The first was constitutional liquation, and eutectic structure consisting of Al_12_Mg_17_ and Mg solid solution existed at the top and upper-middle of the Mg side interface. The second was diffusion reaction, and the two sub-layers of Al_12_Mg_17_ and Al_3_Mg_2_ formed at the lower middle and bottom of the Mg side interface. In addition, the diffusion thickness values of Al_12_Mg_17_ and Al_3_Mg_2_ layers decreased gradually from the lower middle to bottom of the Mg side interface. As the position changes from the middle to the bottom near the Mg side interface, the diffusion coefficient of Al_3_Mg_2_ phase rapidly decreases from 3.14 × 10^−12^ m^2^/s to 6.9 × 10^−13^ m^2^/s and the diffusion coefficient of Al_12_Mg_17_ phase decreases from 6.8 × 10^−13^ m^2^/s to 1.5 × 10^−13^ m^2^/s.

## 1. Introduction

Aluminum (Al) and magnesium (Mg) alloys have been extensively used in automotive, electronics and aerospace industries. The composite structure of Al and Mg alloys has attracted the attention of many researchers because of the advantages of lightweight [1,2]. But combining Al alloy with Mg alloy still face a big challenge. Conventional welding methods, such as fusion welding, are easy to form crack, porosity and a large number of Al-Mg brittle intermetallic compounds (IMCs) due to its high temperature [3,4,5]. So fusion welding of Al and Mg alloys is very difficult used for practical application.

Friction stir welding (FSW), as solid-state joining technology [6], is characterized by the absence of melting, low temperature. Thus, FSW has high possibility of welding Al and Mg alloys compared to fusion welding. Much work has been carried out on the FSW of Al to Mg in butt joint [7,8,9]. Hirano et al. [10] previously reported that dissimilar FSW joints of 5182 Al and AZ31 Mg with a same thickness of 1.2 mm were successfully obtained. The tensile test showed that dissimilar joints have a high joint efficiency of around 77%. Malarizh et al. [11] welded 6 mm-thick plates of 6061 Al and AZ31B Mg at 400 rpm and 19.8 mm/min using a tool with 21 mm shoulder diameter. The joints showed tensile strength of 192 MPa and joint efficiency of 89% in comparison to Mg alloy. However, Mclean et al. [12] reported that joints of 12 mm-thick 5083 Al and AZ31B Mg alloys by FSW were so weak that the sample fractured under the relatively light forces. Moreover, this scanning electron microscopy (SEM) analysis of the welds revealed that the Al_12_Mg_17_ intermetallic compounds (IMC) layer varied in thickness and the debonding had occurred between the IMC layer and Al alloy. The above results suggest that it is more and more difficult to obtain reliable joints with the increase of plate thickness, and the fracture failure may be related to the formation of IMCs.

As known to all, the presence of IMCs is an important issue in dissimilar FSW of Al and Mg alloys because it could not only affect the strength of joints, but also relate to liquation crack [13,14]. Sato et al. [15] and Chen et al. [16] reported that liquid formation occurred rather than solid state diffusion during Al/Mg FSW with thickness less than 6mm, leading to the formation of γ-Al_12_Mg_17_. However, it has been reported that the formation of intermetallics in short welding time is due to the higher diffusion rate in FSW based on the thermal history of FSW [2]. Therefore, the formation of IMCs in FSW of dissimilar metals is still controversial and needs further investigation. On the other hand, there exists a big difference in temperature and microstructure along the thickness direction when thicker plates are butt welded by FSW. For instance, Martinez et al. [17] concluded that a heat gradient distributed along the thickness direction, resulting in the microstructure difference at top and bottom of weld zone for FSW of 13 mm thick Al 7449 alloys. Xu et al. [18] stated that temperature reduced from the top to bottom for FSW of 14 mm thick 2219-O Al alloys. Canaday et al. [19] demonstrated that higher temperature might be located at mid-depth in nugget due to greater heat extraction in FSW with 32 mm thick 7050 Al alloys. The temperature difference along the thickness direction would affect local microstructure. In particular, the formation of IMCs would be affected by the temperature difference in dissimilar FSW of thick plate Al/Mg alloys according to the Al-Mg binary phase diagram, as shown in Figure 1 [20].

Based on the aforementioned investigations, the IMCs of weld interface along thickness direction would vary due to the existence of a temperature difference for thick plate alloys. However, it is unknown that the effect of heat gradient along thickness on interfacial IMCs of dissimilar FSW of Al/Mg alloys with thickness exceeding 12 mm. Thus, the present study examines the features of interfacial IMCs during dissimilar FSW of Al/Mg alloys with 20 mm thick, and discusses its formation mechanism.

## 2. Materials and Methods

The materials used for FSW were 5A06 Al alloy and AZ31B Mg alloy plates (150 × 60 × 20 mm^3^). The chemical compositions of both materials are given in Table 1. The welding specimens are prepared by FSW on an X53K type FSW machine (Tonmac, Nantong, China). Before FSW, the assisted-heating device is used to reduce the temperature difference between the top and bottom of the base materials, and improve the flow ability of plastic material in bottom zone of Al and Mg plates. Through previous experiments at different heating temperatures, the weld shape is better when the heating temperature is about 220 °C. Thus, the heating temperature was set to 220 °C in this experiment.

In order to control the composition of the Al and Mg alloys along thickness in weld zone, an inclined butt joint was designed. Here it was noted that the slope angle of the welded sample and taper of the pin was also prepared as 9°. During the FSW, the Al and Mg alloys were placed at advancing side (AS) and retreating side (RS), respectively, as shown in Figure 2a. The tool cut just 0.5 mm to the Mg alloy side, as shown in Figure 2b. The travel and rotation speed were 23.5 mm/min and 375 rpm, respectively. The tool was rotated clockwise and tilted 2° from the plate normal direction when viewed from above. In order to measure the peak temperature of interface along thickness direction during stable welding process, thermocouples were placed in the position shown in Figure 2b. K-type thermocouples with a diameter of 1.0 mm were embedded in blind hole at depth of 50 mm from the edge of the plates. 

After FSW, microstructure features of the joints were observed on transverse cross section. To obtain microstructure information, the welded samples were etched and unetched, respectively. The unetched sample was used for backscattering electron (BSE) analysis in following procedure. The etched sample was prepared in three steps. Firstly, Al alloy was etched with a Keller reagent for 2.5 min and then wiped with 4% nitric acid solution for 20 s. Secondly, Mg alloy was etched with a solution consisting of 4.2 g picric acid, 10 mL acetic acid, and 10 mL distilled water in 80 mL ethanol for 25 s. The final step was to dip them in a solution of 4 g KMnO_4_ and 2 g NaOH in 100 mL distilled water for 10 s.

Meanwhile, the structural stability of IMCs was investigated by using the thermodynamic calculation method. The microstructure and chemical composition in the weld interface were analyzed by field emission scanning electron microscopy (FESEM, Nova NanoSEM 450, FEI, Hillsboro, OR, USA) with an energy-dispersive spectroscopy (EDS). The chemical composition from EDS was input to JMatPro 7.0 software for predicting the existed phase. The localized phase structure in the weld interface was determined using micro- X-ray diffraction (XRD). Micro-XRD (RIGAKU Rapid IIR, Tokyo, Japan) was performed with Cu Kα radiation at 40 kV and a relatively high tube current of 250 mA. The diameter of X-ray collimator was 0.1 mm.

## 3. Results

### 3.1. Peak Temperature of Al/Mg Joints

Figure 3a shows the thermal cycles measured near the joint interface (marked position A, B, C and D) in dissimilar FSW of Al and Mg alloys. The travel speed and rotation speed are 23.5 mm/min, and 375 rpm, respectively. The peak temperatures of position A and B, located at the Al side interface, are 436.8 °C and 418.5 °C, respectively. The temperature difference between the middle and bottom is about 18.3 °C. On the other side of the weld, the peak temperature of position C and D are 438.8 °C and 424.2 °C, respectively, with a temperature difference of 14.6 °C. Figure 3b shows the magnified thermal cycle at the position C in Figure 3b.

The result mentioned above indicates that temperature distribution along thickness direction is uneven, and exists in similar FSW for thick plate Al alloys [21,22]. It is noteworthy that peak temperature of position C run to 438.8 °C, which slightly exceeds 437 °C that the eutectic reaction ${\mathrm{Mg}+\mathrm{Al}}_{12}{\mathrm{Mg}}_{17}\to L$ would occur. Based on the above analysis, the peak temperature at the top and middle of the plates in this experiment has exceeded the eutectic temperature 437 °C, and liquid films would form along the interface Al and Mg alloys [23]. On the other hand, the weld root performs a relatively low temperature than the top and middle, caused by suffering a smaller stirring force and frictional force than the weld surface.

### 3.2. Macro-Microstructure of the Al/Mg Joints

Figure 4 shows the low magnification overview of dissimilar Al/Mg butt joint by using the travel speed of 23.5 mm/min and rotation speed of 375 rpm. Sound FSW joints without macro-defects are obtained. It is seen from Figure 4 that the multi-layer structures are observed in the weld region, which is apparently different from the FSW of thin plate Al and Mg alloys [15].

On the top of the weld zone, the large dark phase is mixed with a small amount of light grey phase. The dark phase is likely the Mg-rich phase since it is much more susceptible to corrosion and preferentially etched as dark. The light grey phase may be Al-rich phase because of the corrosion resistance of 5A06 Al alloy. The intercalated structure is existed in the middle of the weld zone, also observed at other dissimilar FSW of Al/Mg alloys [24]. It exists apparent onion ring at the bottom of the weld zone, and this phenomenon suggests that materials are incorporated into the tool threads during each rotation, resulting in the formation of lamellae [25,26].

Figure 5 shows the SEM micrographs and EDS maps of the Mg side interface, marked as white dotted rectangles in Figure 4. The interfacial features of different regions (marked as A, B, C, and D) of Mg side interface are discussed as follow. In order to distinguish the phases of Mg side interface, Table 2 presents the chemical composition of the position marked by the number 1 through 14 in Figure 5.

On the left of region A, it consists mainly of light grey phase (position 1 and 2) and black and grey laminar phase (position 3). Quantitative analysis of chemical compositions by EDS shows that the light grey phase consists of about 60.0 wt% Mg and 40.0 wt% Al, while the black and grey laminar phase contains 67.6 wt% Mg and 32.4 wt% Al, as shown in Table 2. This result suggests that the light grey phase is primary Al_12_Mg_17_ and the black and grey phase is a eutectic structure consisting of Al_12_Mg_17_ phase and Mg solid solution. The white and continuous layer is visible near the Mg side interface, as shown at position 4 in Figure 5a. In order to distinguish this white layer, the BSE image is presented at the lower left corner of Figure 5a. It can be seen from the Figure 5a that the white and continuous layer is actually part of the eutectic structure. The reason for the formation of the white and continuous layer is that edge morphology of the Mg side interface is sharp and prominent, which leads to the high reflection intensity of second electrons [27].

It is similar to the top that the upper-middle microstructure also consists of eutectic phase (Mg + Al_12_Mg_17_) and primary Al_12_Mg_17_ phase. Interestingly, the size of the eutectic structure at the upper-middle is smaller than that at the top. This may be ascribed to the lower peak temperature and short duration time in the upper middle than that in the top, so that the amount of melted metals in the upper-middle is less than that in the top, according to the Section 3.1. Moreover, the BSE image of the white and continuous layer at position 4 is indicated at lower left in Figure 5b. It can be seen that white and continuous layer is also a part of eutectic structure.

There exists a significant difference that no eutectic structure is observed in Figure 5c, compared with Figure 5a,b. It is mainly caused by that the temperature of the lower middle is below the eutectic temperature (437 °C). Additionally, the BSE and EDS map in Figure 5c shows that there are two distinct IMC layers formed along the Mg side interface. Quantitative analysis of the chemical compositions of position 8 indicates that this region is the Al solid solution. The position 9 and 10 mainly contain Al and Mg element. From the ratio of Mg to Al the position 9 and 10 should be Al_3_Mg_2_ and Al_12_Mg_17_, respectively. The thickness of the layer Al_12_Mg_17_ is approximately 8.35 μm lower than that of the layer Al_3_Mg_2_, which is ascribed to rapid growth rate of Al_3_Mg_2_ [28]. These findings indicate that the formation of two IMC layers is caused by the diffusion reaction rather than fusion solidification.

It is observed from Figure 5d that the two sub-layers are formed along the Mg side interface. According to the EDS results of position 14, it is deduced that this layer is identified as Al_12_Mg_17_ phase with approximately 3.89 μm thickness. The other layer is identified as Al_3_Mg_2_ phase with thickness of 7.62 μm, which is significantly larger than that of the Al_12_Mg_17_ layer. The same phenomenon was also been reported by Panteli [29] and Lv et al. [30] The straight shape of IMC layer in Figure 5d is obviously different with that in Figure 5a,b, and this straight shape of IMC layer has also been reported in the field of Al and Mg diffusion bonding [31]. This result indicates that the formation of IMC layers in Figure 5d is caused by the diffusion reaction. 

Based on the above analysis, this kind of morphology difference of the IMC layer should be caused by the constitution liquation and the diffusion reaction. The more detailed analysis about the formation of IMC would be carried out in the Section 4.2.

Figure 6 and Figure 7 show the SEM micrographs and EDS maps of the Al side interface, marked as white dotted rectangles in Figure 4, respectively. The interfacial features of different regions (marked as E, F, and G) of Al side interface are discussed as follow. In order to distinguish the phases of Al side interface, Table 3 presents the chemical composition of the position marked by the number 1 through 8. 

As indicated in white arrow in Figure 6a and Figure 7a, there are continuous crack and dispersive eutectic structure existed at the top of the Al side interface comparing with the Figure 5a. This eutectic structure is identified as Mg solid solution and Al_12_Mg_17_ by the EDS analysis. In addition, local magnified map in the Figure 5a clearly indicates that two sub-layers are formed near the top of Al side interface, as shown at position 2 and 3. Here an abnormal phenomenon is observed that the thickness of Al_12_Mg_17_ layer is obviously larger than that of layer Al_3_Mg_2_. This might be due to the fact that existence of cracks prevents Al atoms from diffusing to the interface. 

It is seen from the magnified map in the Figure 6b and Figure 7b that single Al_3_Mg_2_ layer and continuous crack are formed at middle of the Al side interface. As can be seen from Figure 4, the middle of interface is light gray, which indicates that the content of Al is higher than that of Mg. This causes insufficient diffusion of Mg atoms to the Al atoms, which is easy to form Al_3_Mg_2_ at the Al side [32].

Figure 6c shows the bottom microstructure of the Al side interface with respect to the region G in Figure 4. The EDS result indicates that the bottom of Al side interface mainly contains Al substrate and Al_3_Mg_2_. It should be noted that there was no diffusion layer formed at the bottom of Al side interface, as shown in Figure 7c. This might be explained by lower temperature and shearing force, leading to insufficient diffusion and the formation of dispersed granular IMC at the bottom of Al side interface.

### 3.3. Micro-XRD Analysis of the Al/Mg Joints

The micro-XRD spectra obtained from different locations in Figure 4 is indicated in Figure 8. The micro-XRD results from location A in Figure 4 show that Al_12_Mg_17_ phase is formed in the top region of Mg side interface, as shown in Figure 8a. The micro-XRD result from location C in Figure 4, below the middle region of Mg side interface, show that the phase in this region contains Al_3_Mg_2_ and Al_12_Mg_17_. The identical phenomenon is occurred at the bottom of Mg side interface, as shown in Figure 8c. Figure 8d–f show the micro-XRD spectra obtained from locations E, F, and G. The micro-XRD result from location E shows that the top region of Al side interface mainly contains Al, Al_3_Mg_2_ and Al_12_Mg_17_. The micro-XRD results from location F and G show that this region contained Al solid solution and Al_3_Mg_2_.

Combined with the analysis of the mentioned thermal cycle, the presence of the Al_12_Mg_17_ phase in the top region of Al and Mg side interface, such as locations A and E, indicates that eutectic reaction ${\mathrm{Mg}+\mathrm{Al}}_{12}{\mathrm{Mg}}_{17}↔L$ could be taken place in the FSW of thick plate Al and Mg alloys. On the other hand, although the peak temperatures of location C, D, F, and G were lower than the eutectic reaction temperature 437 °C, this would result in the formation of Al_3_Mg_2_ and Al_12_Mg_17_ as a result of enhanced mutual diffusion between Al and Mg atoms undergone high welding temperature and high strain rate plastic deformation in FSW [33].

## 4. Discussion

### 4.1. Thermodynamic Calculation of IMCs

It is well known that the formation of IMCs is related to local composition and temperature. Therefore, the formation of IMCs in non-equilibrium phase is calculated by using the JMatPro software. The EDS results of Mg side interface are presented in Figure 5 and Table 2. Then thermodynamic properties are calculated by the equation of thermodynamics, as expressed in following equations:
(1)$$H=U+\int C_{p}dT$$
(2)G=H−TS
where *H* is the enthalpy, *U* is the formation heat, *C_p_* is the isobaric heat capacity of the specific temperature, *T* is the absolute temperature, *G* is the Gibbs free energy and *S* is the entropy of the specific temperature.

The non-equilibrium phases of location 3 and 13 in Figure 5 are calculated, respectively. The *H*, *S*, and *G* of Al_3_Mg_2_ and Al_12_Mg_17_ are used to investigate the precipitation mechanism of IMCs in the Al/Mg joints. It is shown in Figure 9 that *G* of Al_12_Mg_17_ is lower than that of Al_3_Mg_2_ at the range of 0–500 °C. Hence, Al_12_Mg_17_ phase firstly precipitates than Al_3_Mg_2_ phase. This result is coincided with that reported by Panteli et al. [29].

### 4.2. Formation Discussion of the IMCs

In this study, the top and upper-middle of Mg side interface has been exposed to peak temperature higher than 437 °C during friction stirring according to temperature measurement results. Consequently, constitutional liquation has occurred at the top and upper-middle of the Mg side interface during FSW of thick plate Al and Mg alloys. Diffusion reaction would dominate at the bottom of the Mg side interface since this temperature is lower than eutectic temperature. Similarly, the formation of IMCs at the middle and bottom of the Al side interface is also controlled by diffusion reaction. There are the continuous IMCs layer and partially dispersed eutectic at the top of the Al side interface, which mean that liquation and diffusion reaction have occurred at the top of this interface.

In order to explain clearly the formation mechanism of the constitutional liquation and IMCs, the schematic illustration of IMCs formation is shown in Figure 10. Firstly, the diffusion between Al atoms and Mg atoms would be enhanced under the action of high temperature and serve plastic deformation along the Al and Mg side interfaces. Then the constitutional liquation (${\mathrm{Mg}+\mathrm{Al}}_{12}{\mathrm{Mg}}_{17}\to L$) occur when the reaction temperature arrived to 437 °C depending on the local composition, as indicated in Figure 10a. For the lower-middle and bottom of Mg side interface whose temperature is below the eutectic temperature, the formation and growth of IMCs are induced by the diffusion reaction of Al and Mg atoms. The Al and Mg atoms could diffusion into the Mg and Al alloys during the FSW process, respectively, as indicated in Figure 10b. As the inter-diffusion process between Al and Mg atoms continues, the Al_12_Mg_17_ layer first forms near Mg side interface due to its lower Gibbs free energy, while Al_3_Mg_2_ nucleates near Al side interface as the Mg atoms diffused to Al atoms [34]. The second layer of Al_3_Mg_2_ develops near Al side interface when the Mg atoms diffuse to Al atoms continuously. Although the entire IMCs layer continues to grow, the Al_3_Mg_2_ layer develops at a higher growth rate, becoming thicker than the Al_12_Mg_17_ layer. The results are also consistent with the work of Dietrich et al. [35] and Panteli et al. [29].

Base on the previous study, the relationship between the thickness of IMCs layer and diffusion time could be expressed by using following equations [36]:(3)$$d^{2}=K\cdot t$$where *d* is the diffusion thickness, *t* is the diffusion time, *K* is the diffusion coefficient.

However, the selection of diffusion time in FSW has not been reported as far. In this study, the approximate diffusion time would be discussed in combination with the thermal cycle curve in Figure 2. It is well known that the high temperature range of the stirring area is basically the same as that of the shoulder, and the temperature of the outside shoulder decreases obviously due to the strong heat conduction. Given the shoulder diameter (*D*) and travel speed (*V*), the thermal history time (diffusion time, *t*) of interfacial atoms during Al/Mg FSW process can be expressed as:(4)$$t=\frac{D}{V}$$

The diffusion time (*t*) is about 102 s by substituting shoulder diameter (*D* = 40 mm) and travel speed (*V* = 23.5 mm/min). Then the diffusion coefficient can be calculated according to the diffusion thickness and diffusion time, as shown in Table 4. Compared to the diffusion coefficient obtained by Jin et al. [37], these results are higher. Because of an increased vacancy concentration and dislocation generation caused by severe deformation, the weld interface in Al/Mg FSW might be expected to enhance diffusion. In addition, the diffusion coefficients of Al_3_Mg_2_ and Al_12_Mg_17_ at position C are greater than those at position D. This also confirms that the diffusion coefficient is related to the degree of plastic deformation. As can be seen from Table 4, the measured thickness of Al_3_Mg_2_ and Al_12_Mg_17_ layer is larger than the calculated at position C and D. This might be attributed to the fact that the growth rate of IMCs is higher in severe plastic deformation than that in static condition [29].

## 5. Conclusions

The conclusions are as follows. 

1Dissimilar joints of thick plate 5A06 Al and AZ31B alloys are butt welded using an inclined butt joint by FSW.2There are two different formation mechanisms of intermetallic compounds at the interface of thick plate Al/Mg joints. The one is component liquefaction, the other is reaction diffusion.3As the position changes from the middle to the bottom near the Mg side interface, the diffusion coefficient of Al_3_Mg_2_ phase rapidly decreases from 3.14 × 10^−12^ m^2^/s to 6.9 × 10^−13^ m^2^/s and the diffusion coefficient of Al_12_Mg_17_ phase decreases from 6.8 × 10^−13^ m^2^/s to 1.5 × 10^−13^ m^2^/s.

## Figures and Tables

**Figure 1 materials-12-02661-f001:**
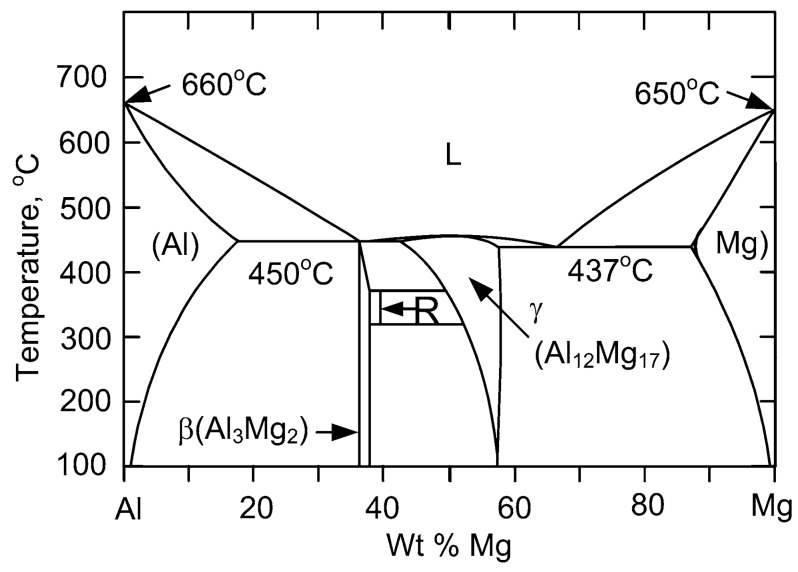
The Al-Mg phase diagram [20].

**Figure 2 materials-12-02661-f002:**
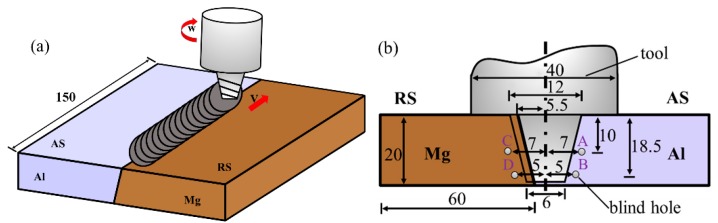
(**a**) Schematic of butt configuration in dissimilar friction stir welding (FSW) Al/Mg alloys (**b**) Temperature measurement (unit: mm).

**Figure 3 materials-12-02661-f003:**
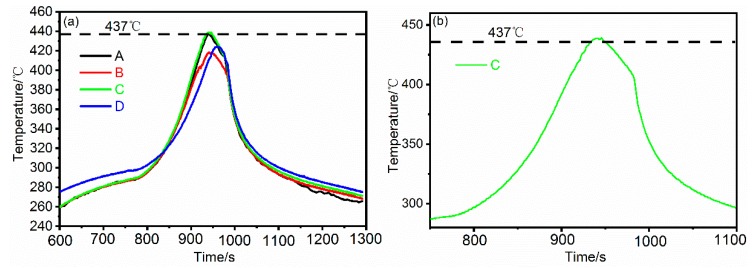
(**a**) The thermal cycle measured at different positions in Figure 2b near the interface of FSW joint. (**b**) The magnified thermal cycle at the position C.

**Figure 4 materials-12-02661-f004:**
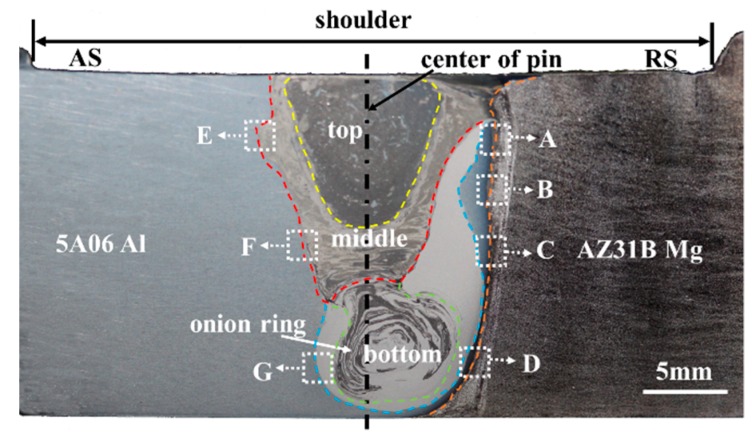
Low magnification overview of dissimilar FSW of Al/Mg alloy.

**Figure 5 materials-12-02661-f005:**
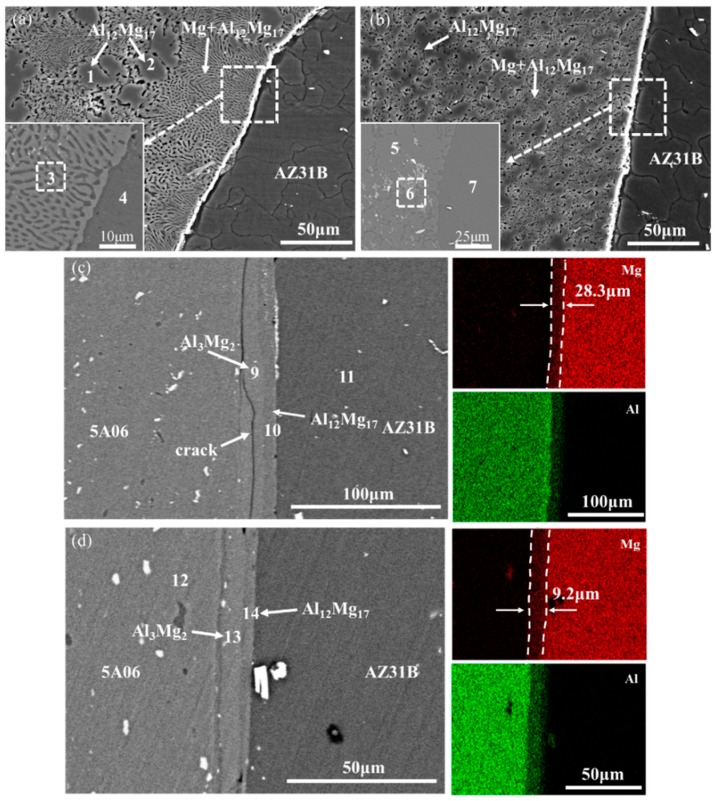
SEM micrographs and EDS maps of the AZ31B Mg alloy interface: (**a**–**d**) represent the location A, B, C and D, respectively.

**Figure 6 materials-12-02661-f006:**
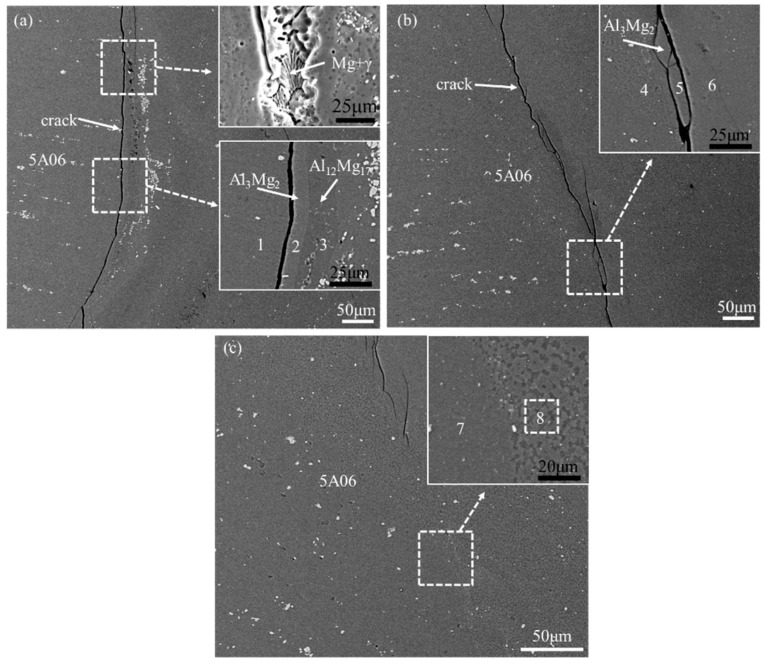
SEM micrograph of the 5A06 Al alloy interface: (**a**–**c**) represent the location E, F and G, respectively.

**Figure 7 materials-12-02661-f007:**
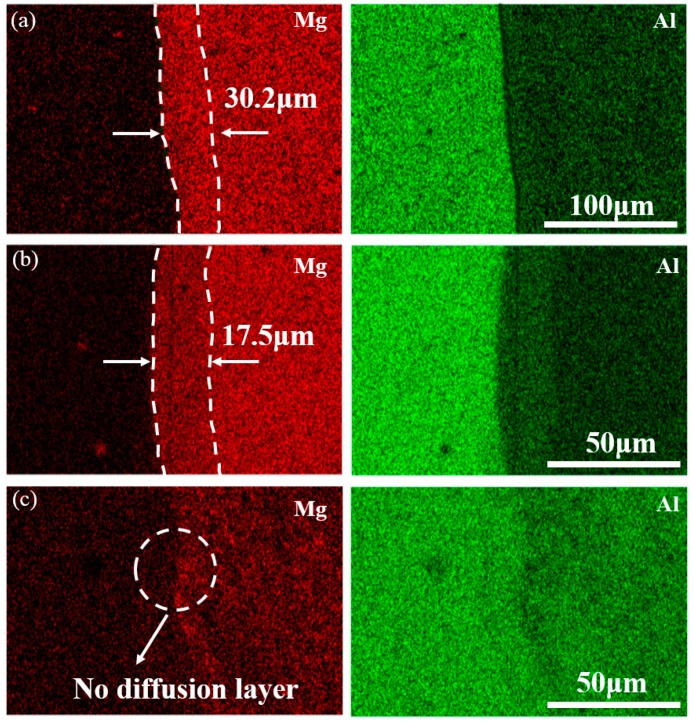
Energy-dispersive spectroscopy (EDS) maps of the 5A06 Al alloy interface: (**a**–**c**) represent the location E, F and G, respectively.

**Figure 8 materials-12-02661-f008:**
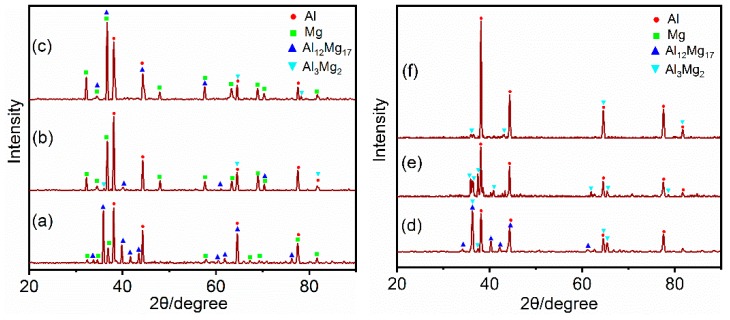
Micro-XRD spectrums from different positions shown in Figure 4: (**a**–**f**) represent the location A, C, D, E, F and G, respectively.

**Figure 9 materials-12-02661-f009:**
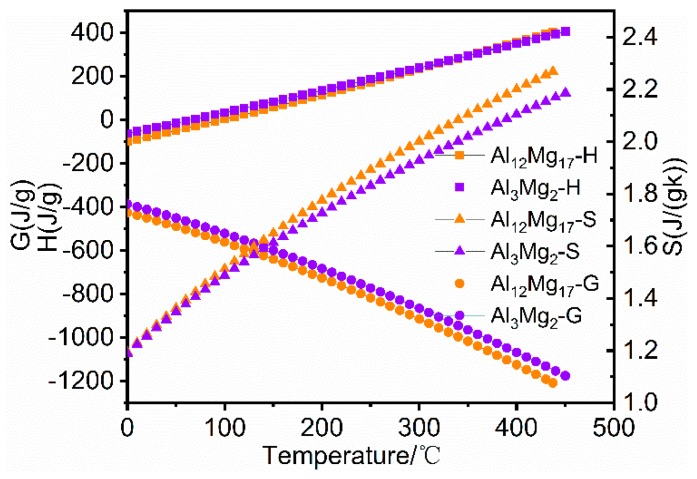
Thermodynamic calculation of Al_3_Mg_2_ and Al_12_Mg_17_ phases.

**Figure 10 materials-12-02661-f010:**
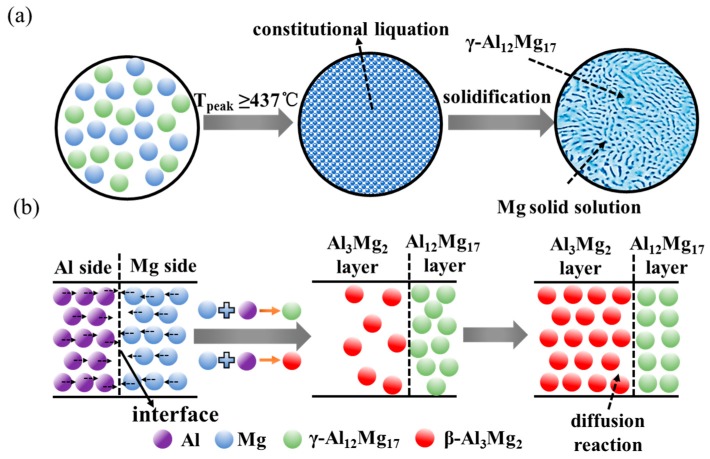
Schematic illustration of the formation of constitutional liquation and reaction diffusion at different position during Al/Mg dissimilar FSW. (**a**) Constitutional liquation at the top of Mg side interface; (**b**) reaction diffusion at the lower-middle and bottom of Mg side interface

**Table 1 materials-12-02661-t001:** The nominal composition of AZ31B magnesium (Mg) and 5A06 aluminum (Al) (in wt.%).

Materials	Al	Zn	Mn	Si	Cu	Fe	Ni	Mg
AZ31B	2.5~3.5	0.6~1.4	0.2~1	≤0.1	≤0.05	≤0.005	≤0.005	Bal.
5A06	Bal.	≤0.2	0.5~0.8	0.4	0.1	≤0.4	0.1	5.8~6.8

**Table 2 materials-12-02661-t002:** Chemical compositions (Weight Percent) measured by EDS at Locations shown in Figure 5.

Location	Mg	Al	Total	Possible Phase(s)	Location	Mg	Al	Total	Possible Phase(s)
1	61.0	39.0	100	Al_12_Mg_17_	8	6.6	93.4	100	Al substrate
2	60.7	39.3	100	Al_12_Mg_17_	9	37.3	62.7	100	Al_3_Mg_2_
3	67.6	32.4	100	Al_12_Mg_17_ + Mg	10	57.6	42.4	100	Al_12_Mg_17_
4	96.5	3.5	100	Mg substrate	11	96.7	3.3	100	Al substrate
5	59.5	40.5	100	Al_12_Mg_17_	12	0	100	100	Al substrate
6	70.0	30.0	100	Mg + Al_12_Mg_17_	13	37.0	63.0	100	Al_3_Mg_2_
7	97.0	3.0	100	Mg substrate	14	60.1	39.9	100	Al_12_Mg_17_

**Table 3 materials-12-02661-t003:** Chemical compositions (Weight Percent) measured by energy-dispersive spectroscopy (EDS) at Locations shown in Figure 6.

Location	Mg	Al	Total	Possible Phase(s)
1	6.9	93.1	100	Mg substrate
2	37.3	62.7	100	Al_3_Mg_2_
3	58.4	41.6	100	Al_12_Mg_17_
4	15.2	84.8	100	Al substrate
5	36.0	64.0	100	Al_3_Mg_2_
6	37.4	62.6	100	Al_3_Mg_2_
7	9.3	90.7	100	Al substrate
8	20.2	79.8	100	Al+ Al_3_Mg_2_

**Table 4 materials-12-02661-t004:** Diffusion coefficients of aluminum (Al)/magnesium (Mg) interface.

Position	IMCs layer	Diffusion Thickness (μm)	Diffusion Time (s)	Diffusion Coefficient (10^−12^ m^2^ s^−1^)
C	Al_3_Mg_2_	17.91	102	3.14
	Al_12_Mg_17_	8.35	102	0.68
D	Al_3_Mg_2_	8.4	102	0.69
	Al_12_Mg_17_	3.89	102	0.15

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
