# Peer review of "Formation Investigation of Intermetallic Compounds of Thick Plate Al/Mg Alloys Joint by Friction Stir Welding"

_materials, 2019, doi:10.3390/ma12172661_

Round 1

Reviewer 1 Report

This article is interesting and may be considered original. 

In order to meet the high quality of journal Materials, a major revision should be suggested. Publication is possible  after solving some errors listed below:

The structure of this paper is not sufficient for journal Materials. Section 2 (Materials and Methods) mixed the description of the experiemnalan methods and their results. The results of experimental investigations should be described in detail in a new chapter named for example Results which followed section 2. This section may be divided by subheadings. It should provide a concise and precise description of the experimental results, their interpretation as well as the experimental conclusions that can be drawn. See Guide for Authors.

The Authors used different notations of alloys used "5A06 Al alloy" and "Al 5A06", "Mg AZ31B" and AZ31B Mg alloy" It should be unified.

Page 1: Type of the paper should be specified.

line 42: What does "r/min" mean? If revolution per minute, in the whole manuscript use "rpm" or "rev./min" insted of "r/min".

line 74. The Authors says that the dimensions of the plates are 150x60x20 mm3. However, in Figure 1 the length os specimen is 100? By the way, in the caption of figure 1 add unit of dimensions.

line 78: "[...] the heating temperature was set to 220℃." It should be explained for the readers why this value of temeprature is assumed.

line 83: Why the tool cut just 0.5 mm to the Mg alloy? Why the welding process was not symmetrical? It should be explained.

            If the specimens were a rectangular prism according to the information in line 74 how technically you assurred the cutting of the 0.5 mm of Mg alloy using conical probe?

line 182: Lv [30] must be changed to Lv et al. [30]

table 2: Is it necessary to present, for each location, the total percentage composition of the Mg/Al joint. Is it possible a situation when a sum of Mg and Al composition is smaller than 100%? See also table 3.

The font style of notations of parameters in the equations and in the text must be unified.

The whole manuscript. Please use space between value of specific parameter and unit, i.e. 21 mm insted of 21mm.

figure 9. What does "k" mean in the unit of S-parameter? Did you mean "K"?

lines 317-318. "This section is not mandatory, but can be added [...]" ?

References list must be formatted according to requirements of publisher.

Although I am not a native English speaker, in my opinion, the manuscript needs to be proofread properly to avoid the some grammatical errors and technical jargon.

Author Response

Thank you very much for reviewer’s comment on our paper "Formation investigation of intermetallic compounds of thick plate Al/Mg alloys joints by friction stir welding" . We have revised the manuscript according to your kind advices and reviewer’s detailed suggestions. Please see the attachment.

Reviewer 2 Report

Its an interesting manuscript about the possible formation of intermetallic compounds Al/Mg alloy joints by friction stir welding.

General comments:

-The title suggests there were different joints, however only one is presented in detail in the manuscript? Is there only one, or were there pleriminary joints, how were the welding parameters determined?

-The mechanic properties of the joints were completely neglected, how these intermetallics influence the joint properties? Are there any tensile or bending test....etc. At least some hardness profiles, could say a lot (compared to literature data of homogeneous joints).

-In conclusion please give a hint for possible industrial usage of the findings, potential window of application, How can a welding engineer (which I am) use the results of this paper?

Some specific remarks, questions, comments are also listed in the manuscript_with_reviewers_comments

With proper corrections I think the manuscript could satisfy the publication criteria in Materials.

Author Response

We have checked the manuscript and revised it according to the comments. Please see the attachment.

Round 2

Reviewer 1 Report

I am comfortable with the modifications made by the authors and I have no further comments. In my opinion this manuscript is now acceptable for publication.

Author Response

Thanks for your comments on our paper.